# Gate-tunable giant nonreciprocal charge transport in noncentrosymmetric oxide interfaces

Daeseong Choe [1,9], Mi-Jin Jin [1,9], Shin-Ik Kim[2], Hyung-Jin Choi[2], Junhyeon Jo [1], Inseon Oh [1], Jungmin Park[1,3], Hosub Jin[4], Hyun Cheol Koo [5,6], Byoung-Chul Min [5,7], Seokmin Hong[5], Hyun-Woo Lee [8], Seung-Hyub Baek[2,7] & Jung-Woo Yoo [1]*

A polar conductor, where inversion symmetry is broken, may exhibit directional propagation of itinerant electrons, i.e., the rightward and leftward currents differ from each other, when time-reversal symmetry is also broken. This potential rectification effect was shown to be very weak due to the fact that the kinetic energy is much higher than the energies associated with symmetry breaking, producing weak perturbations. Here we demonstrate the appearance of giant nonreciprocal charge transport in the conductive oxide interface, $LaAlO_3$/$SrTiO_3$, where the electrons are confined to two-dimensions with low Fermi energy. In addition, the Rashba spin–orbit interaction correlated with the sub-band hierarchy of this system enables a strongly tunable nonreciprocal response by applying a gate voltage. The observed behavior of directional response in $LaAlO_3$/$SrTiO_3$ is associated with comparable energy scales among kinetic energy, spin–orbit interaction, and magnetic field, which inspires a promising route to enhance nonreciprocal response and its functionalities in spin orbitronics.

[1] School of Materials Science and Engineering-Low Dimensional Carbon Materials Center, Ulsan National Institute of Science and Technology, Ulsan 44919, Korea. [2] Center for Electronic Materials, Korea Institute of Science and Technology, Seoul 02792, Korea. [3] Center for Scientific Instrumentation, Division of Scientific Instrumentation & Management, Korea Basic Science Institute, Daejeon 305-806, Korea. [4] Department of Physics, Ulsan National Institute of Science and Technology, Ulsan 44919, Korea. [5] Center for Spintronics, Korea Institute of Science and Technology, Seoul 02792, Korea. [6] KU-KIST Graduate School of Converging Science and Technology, Korea University, Seoul 02841, Korea. [7] Division of Nano and Information Technology, KIST School, Korea University of Science and Technology, Seoul 02792, Korea. [8] Department of Physics, Pohang University of Science and Technology, Pohang 37673, Korea. [9] These authors equally contributed: Daeseong Choe, Mi-Jin Jin. *email: jwyoo@unist.ac.kr

A two-dimensional electron gas (2DEG) confined at the oxide interface, $LaAlO_3/SrTiO_3$ (LAO/STO), has shown various interesting condensed matter phases and rich spin-orbitronic functionalities associated with broken inversion symmetry[1,2]. This noncentrosymmetric 2D conductor exhibits exotic superconductivity with an unconventional order parameter[3–7]. The Rashba-type spin–orbit interaction due to inherent structural asymmetry ties spin and momentum of electrons in the band structure leading to coupled spin and charge transport[8,9]. The strength of the Rashba interaction highly relies on the location of Fermi level as it stiffly increases near the Lifshitz transition, where degenerate multiple $d$ orbitals mix together[10–12]. Thus, applying gate bias can significantly modulate the strength of the Rashba spin–orbit interaction[13–17]. This tunable Rashba interaction provides versatile spin-orbitronic functionalities. The coupling and conversion between spin and charge transport lead to the alternative generation and detection of spin currents, which have been evidenced in various experimental approaches[17–22]. The observed high efficiency of the spin-charge conversion in this system has been attributed to robust spin–orbit interaction and long-lived electronic states[23].

The inherent structural asymmetry in the conductive oxide interface (Fig. 1a) can induce further intriguing transport properties. The directional charge transports have been observed in the systems with broken inversion symmetry, as exemplified by the semiconductor $p$–$n$ junction diodes[24–26]. However, non-centrosymmetric systems do not necessarily exhibit directional responses as the time-reversal symmetry imposes degeneracy between right- and left-mover with opposite spins in an electronic band structure. Applying a magnetic field that breaks time-reversal symmetry lifts this degeneracy by tilting the energy dispersion (Fig. 1b). Then, the second-order correction to the electrical current due to both symmetry-breaking energies results in the directional charge transport[26]. Rikken et al.[24] provided a phenomenological description for the directional transport in the noncentrosymmetric system by generalizing Onsager's theorem as follows

$$R(I, B) = R_0(1 + \beta B^2 + \gamma IB) \tag{1}$$

where, $R_0$, $I$, and $B$ represent the resistance at zero magnetic field, the electric current, and magnetic field, respectively. $\beta$ is a coefficient of the normal magnetoresistance (MR) and $\gamma$ is a coefficient tensor representing the magnitude of the nonreciprocal resistance. For a polar system, $\gamma$ exhibits maximum magnitude when the polarization $\mathbf{P}$, magnetic field $\mathbf{B}$, and the current $\mathbf{I}$ are orthogonal to each other. Then, the nonreciprocal resistance can be written as

$$\Delta R = R(I) - R(-I) \propto \mathbf{I} \cdot (\mathbf{P} \times \mathbf{B}) \tag{2}$$

Several noncentrosymmetric systems were found to exhibit this magnetochiral anisotropy, such as Bi helix[24], Si FET[27], molecular conductor[28], and BiTeBr[26]. The measured coefficient $\gamma$ is typically very small with an order of $10^{-1}$ to $10^{-4}$ $T^{-1} A^{-1}$. This is because the energy scale of the spin–orbit interaction and the applied magnetic field are generally much smaller than the kinetic energy. Recently, it was reported that this nonlinear effect could be strongly enhanced when the system enters a superconducting state where the electron pairing energy could be strongly disturbed by symmetry-breaking energies[29–31]. The nonreciprocal charge transport could be an alternative probe to assess the strength of the Rashba spin–orbit interaction[26] and could also be utilized for the determination of surface spin textures[32,33]. The directional charge transports have also been observed in various heavy-metal/ferromagnet (HM/FM) bilayer systems[34–37]. In these cases, the spin accumulation via either the spin Hall or the Edelstein effect in combination with spin-dependent scatterings

at the interface was attributed to the origin of the observed unidirectional MRs. The observed ratios of resistance changes ($\Delta R/R$) in HM/FM systems were typically much <1%. The studies of directional charge transports have been focused on metallic systems, where relatively high Fermi energy limits the perturbation from spin–orbit interaction and Zeeman energy. Achieving high magnitude of the directional charge transport and its tunability will be an important foundation toward two-terminal spin-orbitronic devices.

In this study, we report giant nonreciprocal responses in the LAO/STO interface, an archetype of conductive oxide interfaces. The directional responses are measured with both the static and dynamic electrical field, respectively, which show consistent results with each other. In particular, the Rashba spin–orbit interaction in the LAO/STO interface enables us to tune the strength of the nonreciprocal charge transport by applying a gate voltage. Upon increasing a gate voltage, the ratio of resistance change between the rightward and leftward currents is increased up to 2.7%. The coefficient $\gamma$ representing the strength of the magnetochiral anisotropy is measured to be as high as $\sim10^2$ $T^{-1} A^{-1}$, which is about three orders of magnitude higher than those estimated for typical noncentrosymmetric conductors. Moreover, the magnitude of the directional response exhibits additional higher-order magnetic field dependence. The observed behavior of giant directional response in this system is due to the fact that energies associated with the broken inversion symmetry and the broken time-reversal symmetry are comparable to the Fermi energy, which opens an effective route to enhance nonreciprocal responses in polar materials.

## Results

**Nonreciprocal responses upon applying DC currents.** Figure 1c displays a scanning electron microscopy image of the studied Hall bar device, which has a dimension of 15 μm channel width. All results displayed in the main text were measured from two devices (device A and device B), which have the same geometry as shown in Fig. 1c. The devices were fabricated on $5 \times 5$ mm (001) STO substrates. The LAO/STO samples used for our devices have 10 unit cells (4 nm) of LAO layers grown by the pulsed laser deposition (PLD). Fabrication of a Hall bar and contacts to the conducting interface was done following the method used in our previous study of the nonlocal spin diffusion[19]. Details of sample growth and device fabrication were described in the Methods and Supplementary Fig. 1. Before the electrical characterization of the studied devices, we first confirmed electrical contacts to the conductive interface underneath the LAO barrier through $I$–$V$ curves as shown in Supplementary Fig. 2. The temperature dependence of the channel resistance is displayed in Supplementary Fig. 3, which follows general metallic behavior of the LAO/STO interface[1]. The temperature dependence of the mobility and the carrier concentration obtained by the Hall effect measurement are also displayed in Supplementary Fig. 4. The measured sheet resistance ($R_{sq}$) was $\sim426$ Ω at 2 K. The sheet carrier density ($n_s$) was $\sim1.56 \times 10^{13}$ $cm^{-2}$ at 2 K. Based on the 2DEG, the Fermi energy can be obtained by $\varepsilon_F = \frac{\hbar^2}{2m^*}(2\pi n_s)^2 \sim 18.7$ meV (where $m^* \approx 2m_e$)[38,39]. We note that this value is within one order of magnitude of the spin splitting energy in LAO/STO. The estimated momentum scattering time and characteristic mean free path are $\tau = \frac{m^* \sigma_s}{n_s e^2} \sim 1.07$ ps and $\lambda_e = v_F \tau \sim 61.2$ nm, respectively. Figure 1d displays $R_{xx}$ measured for $I_x = +30$ μA and $I_x = -30$ μA at 8 K upon varying applied magnetic field $B_y$. Measurements were done with a magnetic field sweeping rate of 10 mT $s^{-1}$. Large negative MR upon increasing $B_y$ was observed. This negative in-plane MR in LAO/STO was attributed to the anisotropic deformation of the Fermi surface upon increasing Zeeman energy, which results in

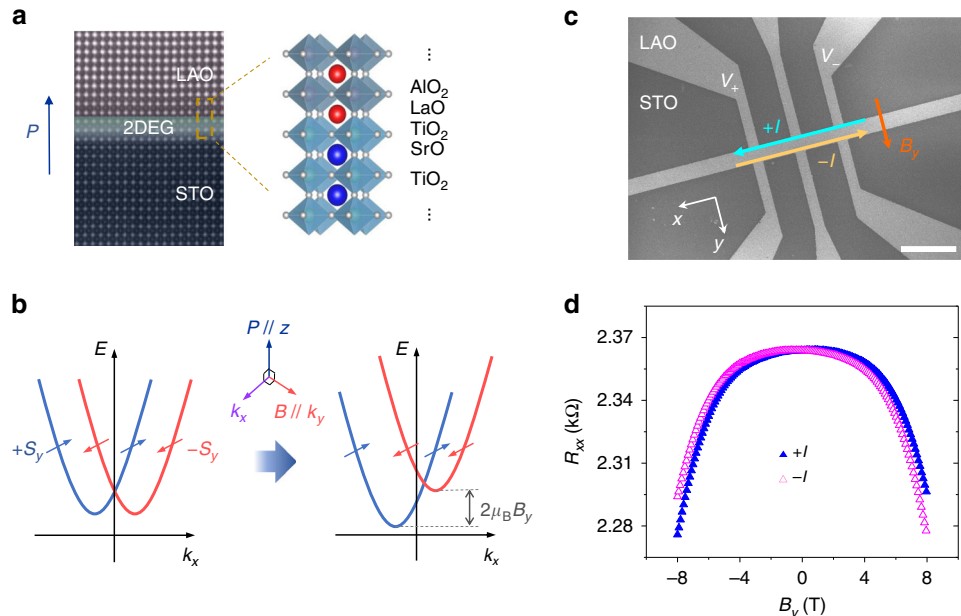

**Fig. 1** Nonreciprocal charge transport in the noncentrosymmetric LAO/STO conductive interface. **a** Transmission electron microscopy image and crystal structure of the LAO/STO displaying conductive interface with the broken inversion symmetry. **b** Electron dispersion in the presence of the Rashba spin–orbit interaction and the external magnetic field. Nonreciprocal charge transport arises when the polarization **P**, magnetic field **B**, and the current **I** are orthogonal to each other. **c** Scanning electron microscopy image of the studied Hall bar device (device A) fabricated on a $5 \times 5$ mm (001) STO substrate. Channel width was 15 μm. Length of scale bar is 50 μm. **d** $R_{xx}$ measured for both direction of currents $+I_x$ and $-I_x$, while sweeping the applied magnetic field $B_y$. Measurements were done at 8 K with $I_x = 30$ μA. Results clearly exhibit the presence of nonreciprocal response in the LAO/STO interface

suppressed interband scattering and reduced sheet resistance[40]. In particular, a significant difference between $R_{xx}(+I_x)$ and $R_{xx}(-I_x)$ was clearly observed, indicating the existence of the nonreciprocal charge transport in the LAO/STO interface. The measured ratio of resistance change, $\frac{\Delta R_{xx}}{R_{xx}} = \frac{\left(R_{xx}\left(+I_x,B_y\right) - R_{xx}\left(-I_x,B_y\right)\right)}{1/2 \cdot \left(R_{xx}\left(+I_x,B_y\right) + R_{xx}\left(-I_x,B_y\right)\right)}$ is very large compared with those measured for other noncentrosymmetric conductors[24,26–28].

As the Rashba spin–orbit interaction in the LAO/STO conductive interface is gate-controllable, a further tuning of the nonreciprocal response can be achieved by applying a back-gate voltage $V_g$, as shown in Fig. 2a, b. The measurements were done with an applied current $I_x = \pm 30$ μA at $T = 8$ K. Figure 2a displays MR, defined as MR $= (R_{xx}(B) - R_{xx}(B=0))/R_{xx}(B=0)$, measured at various applied gate voltages. For each applied gate voltage, MR was measured for both direction of currents $+I_x$ and $-I_x$, while sweeping the applied magnetic field $B_y$. The ratio of resistance change $\Delta R_{xx}/R_{xx}$ in between $+I_x$ and $-I_x$ estimated at $B_y = 8$ T is plotted as a function of gate voltage in Fig. 2b. The strong asymmetric $V_g$ dependence of the nonreciprocal response can be observed. The increase of the Rashba spin–orbit interaction and electron accumulation by raising $V_g$ in the LAO/STO interface were known in previous reports[10–13]. As a result of the sub-band hierarchy of the 2DEG along (001) STO (see the inset of Fig. 2b), the electrostatic modulation of orbital occupancy by applying $V_g$ is anticipated to lead a significant variation of the spin–orbit interaction. At $V_g < 0$ V, only $d_{xy}$ orbitals are populated. In contrast, as $V_g$ increases to the regime of accumulation, $d_{xz/yz}$ bands start to be filled, leading to the Lifshitz transition. Then, the spin–orbit interaction starts to increase significantly due to the mixing of multiple $d$ orbitals. According to the literature by Joshua et al.[10], typical quantitative values of the carrier density for the Lifshitz transition were $1.68 \pm 0.18 \times 10^{13}$ cm$^{-2}$. This value corresponds to the Fermi energy of

~20.1 meV within the 2D free electron model. In our studied sample, the obtained value of $n_s$ at $V_g = 0$ and 8 K is ~$1.61 \times 10^{13}$ cm$^{-2}$ (for device B, Supplementary Fig. 4). This value is slightly less than the Lifshitz point. Thus, the Lifshitz transition could occur with increasing $V_g$ across zero voltage. The gate-tunable spin–orbit interaction can be also inferred from the crossover between weak localization and weak anti-localization behaviors at low temperature[13,15,41,42]. The LAO/STO samples in this study also display the crossover between weak localization and weak anti-localization, depending on the applied gate voltage (Supplementary Fig. 5). Further analyses of out-of-plane magnetoconductance curves within a Maekawa–Fukuyama theory were discussed in Supplementary Note 1 and Supplementary Fig. 6, 7, and 8. Results showed that the strong enhancement of the Rashba spin–orbit interaction across the Lifshitz point (Supplementary Fig. 7a) is consistent with previous reports[13].

Although the nonreciprocal charge transport significantly depends on the gate-tunable Rashba spin–orbit interaction, it also highly relies on the carrier concentration. Ideue et al.[26] derived $n_s^{-3}$ dependence of the nonreciprocal response in a single-band 2D Rashba system by using the Boltzmann equation. The carrier concentration of the studied LAO/STO system exhibits a gradual increase with increasing $V_g$ but its variation is very weak (Supplementary Fig. 7d). In contrast, the estimated Rashba spin splitting energy is significantly enhanced with increasing $V_g$ (Supplementary Fig. 7a). Therefore, gate-tuned Rashba interaction mainly accounts for the observed $V_g$ dependence of the nonreciprocal response in this system. As shown in Fig. 2b, the observed nonreciprocal response is nearly negligible for $V_g < 0$ V, whereas it stiffly increases upon applying a positive $V_g$, consistent with the Lifshitz transition across zero gate voltage. Interestingly, the negative in-plane MR also increases significantly with applying a positive $V_g$ and can be collapsed into a single curve by rescaling of the magnetic field $B \rightarrow B/B^*$ ($B^*$ is a

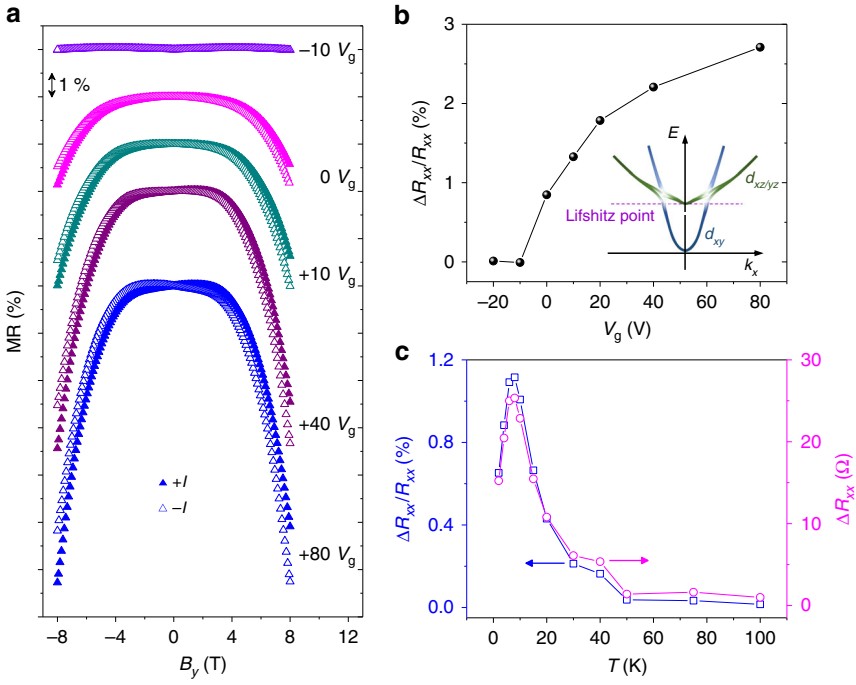

**Fig. 2** Nonreciprocal charge transport in the LAO/STO interface under the DC electrical field. **a** In-plane MR curves measured for the device A at various applied gate voltages. The measurements were done for both $I_x = +30\,\mu A$ and $I_x = -30\,\mu A$ at $T = 8\,K$. The nonreciprocal resistance was strongly enhanced upon increasing gate voltage. The MR curves are shifted vertically for clarity. **b** The nonreciprocal resistance $\Delta R_{xx}/R_{xx}$ as a function of gate voltage. $\Delta R_{xx}/R_{xx}$ was estimated at $T = 8\,K$ and $B_y = 8\,T$ with $I_x = \pm 30\,\mu A$. $\Delta R_{xx}/R_{xx}$ is nearly negligible at $V_g < 0\,V$. $\Delta R_{xx}/R_{xx}$ stiffly increases upon applying positive $V_g$. The inset displays the sub-band hierarchy of the 2DEG at (001) LAO/STO interface. **c** The temperature dependence of $\Delta R_{xx}/R_{xx}$ (left) and $\Delta R_{xx}$ (right) measured at $B_y = 8\,T$ with $I_x = \pm 30\,\mu A$. $\Delta R_{xx}/R_{xx}$ showed the maximum value at around 10 K and nearly disappeared over 100 K

density-dependent value)[40] (see Supplementary Fig. 9). Mechanisms of the nonreciprocal charge transport and negative in-plane MR could be highly correlated, because both effects depend on the anisotropic deformation of a Fermi surface.

The LAO/STO system presents a gigantic nonreciprocal response displaying $\Delta R_{xx}/R_{xx}$ up to 2.7%. The corresponding value of $\gamma$, which characterizes the magnitude of the nonreciprocal response is estimated to be $\gamma \sim 10^2\,A^{-1}\,T^{-1}$. This value is much higher than those observed in other polar conductors. For examples, $\gamma \sim 10^{-3}\,A^{-1}\,T^{-1}$ for a Bi helix[24], $\gamma \sim 10^{-2}\,A^{-1}\,T^{-1}$ for a chiral organic conductor[28], $\gamma \sim 10^{-1}\,A^{-1}\,T^{-1}$ for a 2DEG in Si FET[27], and $\gamma \sim 1\,A^{-1}\,T^{-1}$ for a BiTeBr[26]. If we convert the coefficient $\gamma$ for a current density, then the magnitude of the normalized $\gamma' = \gamma A$ ($A = L_y L_z$ is the cross-sectional area of the sample) becomes as high as $\sim 1.17 \times 10^{-11}\,A^{-1}\,T^{-1}\,m^2$ (thickness was assumed to be $\sim 7\,nm$)[43]. This value is higher than that observed in BiTeBr[26] by more than one order of magnitude.

Figure 2c displays the temperature dependence of the nonreciprocal response. Measurements were done with an applied current of $I_x = 30\,\mu A$ at $B_y = 8\,T$. The estimated nonreciprocal resistance $\Delta R_{xx}/R_{xx}$ showed the maximum value at around 8 K and nearly disappeared over 100 K. The temperature-dependent MR curves were also displayed in Supplementary Figs. 10 and 11. In general, the conductivity of the LAO/STO interface decreases below 10 K due to other quantum corrections, i.e., Kondo effect[44] and weak localization[13]. The LAO/STO samples in our study also displayed such quantum corrections to the diffusive transport at low temperature (see Supplementary Fig. 3b). These effects could be detrimental to the nonreciprocal response at very low temperature below 10 K. Another possible explanation is the slight decrease of a carrier concentration when the temperature is lowered below 10 K (see

the inset of Supplementary Fig. 4b). The decrease of a carrier concentration away from the Lifshitz transition reduces the spin–orbit interaction, so does the nonreciprocal response.

**Nonreciprocal responses upon applying AC currents**. The nonreciprocal charge transport can be further clearly observed upon applying AC input current ($I = I_0 \sin \omega t$) with the lock-in technique. In LAO/STO, the polarization points out-of-plane. If the magnetic field is rotated in the $xy$ plane with an angle $\theta$ between $+x$ axis and the magnetic field, the nonreciprocal voltage from Eqs. (1) and (2) can be written as follows[26],

$$
\begin{aligned}
V^{2\omega}(t) &= \gamma R_0 B I_0 \sin \theta \sin \omega t \times I_0 \sin \omega t = \gamma R_0 B I_0^2 \sin \theta \sin^2 \omega t \\
&= \frac{1}{2}\gamma R_0 B I_0^2 \sin \theta \left\{ 1 + \sin\left(2\omega t - \frac{\pi}{2}\right) \right\}
\end{aligned}
\tag{3}
$$

By measuring out-of-phase components of the second harmonic, we can directly probe the nonreciprocal resistance as follows[26],

$$
\Delta R^{2\omega} = \frac{V_y^{2\omega}}{I_0} = \frac{1}{2}\gamma R_0 B I_0 \sin \theta
\tag{4}
$$

We have investigated the field-angle dependence of the nonreciprocal resistance in response to the dynamic electric field with the device B. Figure 3a illustrates angle-dependent AC measurement with the definition of the coordinates and rotation planes. The zero angles are at $+x$, $+z$, and $+z$, and the directions of rotations are $x$ to $y$, $z$ to $y$, and $z$ to $x$ for $xy$, $zy$, and $zx$ rotations, respectively. The applied magnetic field was rotated in each orthogonal plane, while the second-harmonic resistance $R^{2\omega}$ was recorded for AC input current of $I_{ac} = 200\,\mu A$. Figure 3b displays $R^{2\omega}$ as a function of the $xy$ angle measured for the

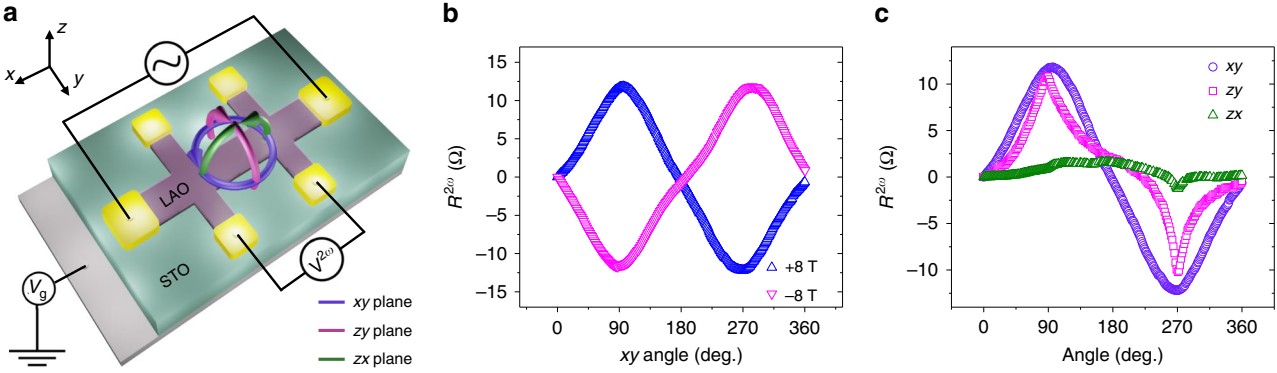

**Fig. 3** Angular dependence of the nonreciprocal charge transport under the AC electrical field. **a** An experimental setup for the measurement of angular-dependent $R^{2\omega}$ and definitions of rotation planes. The zero angles are at $+x$, $+z$, and $+z$, and the directions of rotations are $x$ to $y$, $z$ to $y$, and $z$ to $x$ for $xy$, $zy$, and $zx$ rotations, respectively. **b** $R^{2\omega}$ as a function of the $xy$ angle measured with applied magnetic fields of $B = +8$ T and $B = -8$ T, respectively. Measurements were done with $I_{ac} = 200\ \mu A$ at 8 K. $R^{2\omega}$ displays the maximum/minimum values when the polarization **P**, magnetic field **B**, and the current **I** are orthogonal to each other. **c** $R^{2\omega}$ in response to the applied magnetic field (8 T) rotated in three orthogonal plans ($xy$, $zy$, $zx$). Measurements were done with $I_{ac} = 200\ \mu A$ at 8 K

reversed magnetic field of $B = +8$ T and $B = -8$ T, respectively. $R^{2\omega}$ exhibited maximum magnitude at $\theta = 90°$ and $270°$ with reversed sign for opposite magnetic field. The results showed the relation of $R^{2\omega}(I, B) = -R^{2\omega}(I, -B)$, demonstrating a unidirectional characteristic of MR, i.e., odd to the polarity of the magnetic field. The observed $R^{2\omega}$ signal is not associated with thermoelectric artifacts arising from the vertical temperature gradient (see Supplementary Fig. 12). Figure 3c displays $R^{2\omega}$ in response to the applied magnetic field of 8 T rotated in three orthogonal plans ($xy$, $zy$, $zx$). The magnitude of $R^{2\omega}$ relied on the $y$ component of the applied magnetic field and became nearly negligible when the applied magnetic field was rotated along the $zx$ plane. Here we note that the field-angle dependence of $R^{2\omega}$ in the $zy$ plane is different from that in the $xy$ plane, which will be discussed later in detail. The observed $R^{2\omega}$ at fixed magnetic field direction also showed the dependence on the $y$ component of the applied magnetic field (Supplementary Fig. 13). The first-harmonic $R^{\omega}$ exhibited conventional anisotropic behavior, i.e., a sinusoidal angular dependence with a period of 180° (see Supplementary Fig. 14). Thus, $R^{\omega}$ is invariant under the reversal of the magnetic field.

The gate-dependent nonreciprocal response can also be observed by measuring $R^{2\omega}$. Figure 4a displays $R^{2\omega}$ as a function of the $xy$ angle measured with various applied gate voltages. This result is summarized in Fig. 4b, where the magnitude of the maximum $R^{2\omega}$ (at $\theta = 90°$) is plotted as a function of $V_g$. Results display a strong asymmetric $V_g$ dependence of $\Delta R^{2\omega}$, consistent with DC measurements shown in Fig. 2b. As the nonreciprocal response $R^{2\omega}$ is directly associated with the strength of the Rashba spin–orbit interaction, the sudden increase of $R^{2\omega}$ could be regarded as the signature of the Lifshitz transition in this LAO/STO system.

Ideue et al.[26] derived the second-order current $\left(J_x^{2nd}\right)$ responsible to the nonreciprocal charge transport using the Boltzmann equation, as follows

$$J_x^{2nd} = E_x^2 \left( a \left( \frac{B_y}{\lambda} \right) + b \left( \frac{B_y}{\lambda} \right)^3 + O\left( \left( \frac{B_y}{\lambda} \right)^5 \right) \right) \quad (5)$$

where $E_x$ is the applied electric field and $\lambda$ is the magnitude of the spin–orbit interaction. Both parameters $a$ and $b$, which determine the strength of the nonreciprocal response, are functions of $\mathcal{F}\left( \frac{1}{\varepsilon_F - m\lambda^2} \right)$ ($\varepsilon_F$ is Fermi energy and $m\lambda^2$ is the Rashba splitting

energy)[26]. Thus, the relative strength of the Rashba spin–orbit interaction in comparison with the Fermi energy is a governing factor to produce the nonreciprocal charge transport. This is conceivable, as the nonreciprocal charge transport is in principle a perturbation effect by the spin–orbit interaction. The second-order current in Eq. (5) is odd to the applied $B_y$, i.e., $J_x^{2nd}(-B_y) = -J_x^{2nd}(B_y)$. The electrical conductivity from the second-order current also depends on the direction of the electric field and thus describes the nonreciprocal electric transport as $\sigma^{2nd}(+I_x) = -\sigma^{2nd}(-I_x)$, which is linear to the applied $E_x$. Figure 4c shows $R^{2\omega}$ as a function of the $xy$ angle measured with various applied currents. An excellent linear relationship between the maximum $R^{2\omega}$ and an applied current can be observed as shown in Fig. 4d. The ratio of resistance change $\Delta R_{xx}/R_{xx}$ in response to a DC current also displays a linear behavior as shown in Supplementary Fig. 15.

Figure 4e displays the angle dependence of $R^{2\omega}$ measured with varying applied magnetic field. The observed maximum $R^{2\omega}$ is plotted as a function of the applied magnetic field in Fig. 4f. The maximum $R^{2\omega}$ shows not only a linear dependence on $B_y$ but also a significant higher-order dependence on $B_y$ when the applied field was increased over several Tesla. The appearance of higher-order terms would be associated with a relative strength between $B_y$ and $\lambda$. Following Eq. (5), higher-order dependence terms are negligible when $B_y \ll \lambda$. In LAO/STO, the estimated $m\lambda^2$ values were typically around $\sim 3$ meV in both theoretical[45] and experimental[13,17] studies. The corresponding Zeeman energy $\mu_B B_y$ is $\sim 51.8$ T (assuming $g = 2$). Thus, the nonreciprocal charge transport in LAO/STO may exhibit a higher-order dependence for applied magnetic fields over several Tesla. Applying a gate voltage, which tunes the Rashba spin–orbit interaction, significantly modifies the magnetic field dependence of $R^{2\omega}$ as shown in Supplementary Fig. 16. This nonlinear variation of the non-reciprocal resistance can also be observed in $\Delta R_{xx}/R_{xx}$ measured with a DC current (Supplementary Fig. 17). The higher-order dependence on the applied magnetic field also reflects on the different behavior of $R^{2\omega}$ in between $xy$- and $zy$-plane rotations (shown in Fig. 3c). When the magnetic field is rotated in the $xy$ plane, the direction of the field is always orthogonal to the direction of the polarization. As $\Delta R \propto I \cdot (P \times B)$, $R^{2\omega}$ displays sinusoidal behavior for the rotation of the magnetic field in the $xy$ plane. On the other hand, when the field is rotated in the $zy$ plane, the orthogonal component of the field to the direction of the

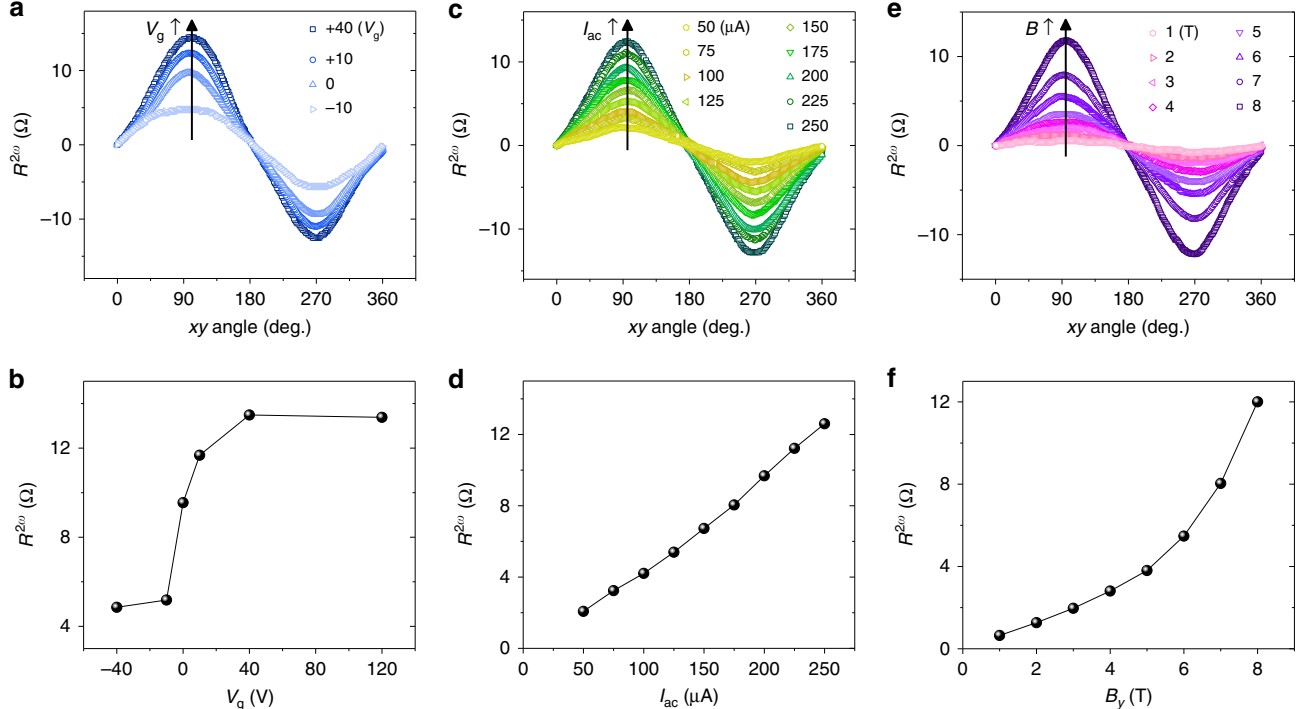

**Fig. 4** Nonreciprocal response $R^{2\omega}$ depending on the applied gate voltage, current, and magnetic field. **a** $R^{2\omega}$ as a function of the $xy$ angle of the applied magnetic field 8 T measured with various gate voltages. Measurements were done with $I_{ac} = 200\,\mu A$ at 8 K. **b** Maximum $R^{2\omega}$ as a function of the applied $V_g$. Results are consistent with DC measurements displaying asymmetric $V_g$ dependence. **c** $R^{2\omega}$ as a function of the $xy$ angle of the applied magnetic field 8 T measured at 8 K with increasing $I_{ac}$. **d** Plot of maximum $R^{2\omega}$ vs. $I_{ac}$ exhibiting linear behavior. **e** $R^{2\omega}$ as a function of the $xy$ angle of the applied magnetic field. Measurements were done at 8 K with $I_{ac} = 200\,\mu A$ for various applied magnetic fields. **f** Maximum $R^{2\omega}$ as a function of the applied magnetic field. $R^{2\omega}$ shows initially a linear dependence on the applied magnetic field and a higher-order dependence at high magnetic field

polarization is not constant and varies as $B \cdot \sin\theta$. As the $\Delta R$ has additional higher-order dependences on $B$ at a high magnetic field regime, the variation of $R^{2\omega}$ becomes more significant at high fields, making a sharp increase of $R^{2\omega}$ near 90° and 270° in the $zy$ plane (Fig. 3c). At a relatively low magnetic field, where $\Delta R$ is linear to $B$, $R^{2\omega}$ displays sinusoidal behavior for the rotation of the magnetic field in the $zy$ plane (see Supplementary Fig. 18).

## Discussion

We showed robust nonreciprocal charge transport in the inversion symmetry-broken LAO/STO conductive interface. The observed nonreciprocal response is highly tunable by applying gate voltages, as the Rashba spin–orbit interaction in the LAO/STO system varies according to the location of Fermi level. The observed value of $\gamma$ was as high as $\sim 10^2\,T^{-1}\,A^{-1}$, which is the largest value reported for noncentrosymmetric conductors. The large magnitude of the nonreciprocal response is due to the fact that the Fermi energy in LAO/STO is relatively low ($\sim 18.7$ meV, estimated value from the device B). Thus, the spin splitting energy ($\sim 3$ meV) arising from broken inversion symmetry is comparable to the kinetic energy producing strong perturbations to the charge transport. We note that the reported nonreciprocal charge transport in strong Rashba interaction systems, such as BiTeBr ($m\lambda^2 \sim 75$ meV), is much weaker than that observed in LAO/STO[26]. Thus, the important factor that produces the large magnitude nonreciprocal charge transport is not just a strong Rashba spin–orbit interaction but the relative strength of the Rashba spin splitting energy to the Fermi energy. The reported strong spin-charge conversion in LAO/STO[17–22] could be also partly due to this aspect. In addition, the observed directional response in LAO/STO exhibits additional higher-order dependence on the

applied magnetic field when the Zeeman energy is not ignorable compared with the Rashba spin splitting energy. In short, the overall behavior of the nonreciprocal charge transport in the LAO/STO interface can be attributed to the comparable energy scales among kinetic energy, spin splitting energy due to broken inversion symmetry, and Zeeman energy due to time-reversal symmetry breaking. The observed behavior of the giant directional response in our study inspires a promising channel to enhanced nonreciprocal charge transport and confirms conductive oxide interfaces as outstanding 2DEG platforms for functional two-terminal devices in spin orbitronics.

## Methods

**Sample growth.** The (001) STO single-crystal substrates ($5 \times 5$ mm) were etched by using buffered hydrofluoric acid and then were annealed at 1000 °C under an oxygen atmosphere to create the $TiO_2$-terminated surface with the clean step-terrace structure. Epitaxial LAO films on $TiO_2$-terminated STO surface were deposited by using a PLD with a KrF excimer laser ($\lambda = 248$ nm) under 1 mTorr $O_2$ pressure. A KrF excimer laser beam was used for a PLD deposition with an energy density of $1.5\,J\,cm^{-2}$ and a frequency of 2 Hz.

**Device fabrication.** The studied devices were fabricated by using both electron beam lithography and photolithography. Each device fabrication step is illustrated in Supplementary Fig. 1. Negative electron beam lithography was first used to create contact patterns. A thermally deposited Al layer (100 nm) was used as a buffer layer for reactive ion (RI) etching of the contact patterns. Then, Ti (10 nm)/Au (20 nm) layers were deposited to fill the openings of the contact patterns left by the etching, followed by the removal of the Al buffer layer. Positive photolithography was then used to pattern the eight-contact Hall bar geometry of the device. An Al buffer layer (100 nm) was used again for the second RI etching. Then, oxygen annealing was performed at 300 °C to anneal out oxygen vacancies created in the STO surface during the etching process. After removing the Al buffer layer, contact electrodes (Ti (5 nm)/Au (60 nm)) were deposited on top of the contact patterns using a shadow mask. The dimension of the studied Hall bar pattern has a

width of $w = 15\,\mu m$ and a length of $l = 40\,\mu m$ (a distance between the two Hall probes).

**Measurement**. All electrical measurements were performed in a Quantum Design Physical Property Measurement System with a horizontal sample rotator. Electrical contacts to Au pads and a back-gate of the device were made with Ag paste using copper wires. The four-terminal $\Delta R_{xx}$ was measured using a Keithley 2636 sourcemeter and a Keithley 2182 nanovoltmeter. For AC measurements, the AC current of 10 Hz was supplied by a Keithley 6221 current source and both the first- and second-harmonic signals of the AC voltage were measured by a SR830 DSP of Stanford Research Systems. The back-gate voltage was applied by using the Keithley 2636. For $V^{\omega}$, in-phase components were recorded, whereas out-of-phase $(-\pi/2)$ components were recorded for $V^{2\omega}$. All $R^{2\omega}$ (or $R^{\omega}$) signals shown in the main text and Supplementary Information are out-of-phase (or in-phase) components of the lock-in measurement.

## Data availability
The data that support the findings of this study are available from the corresponding author upon request.

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

## Acknowledgements
This work was supported by the National Research Foundation of Korea (NRF) grant funded by the Korea government (2017R1A2B4008286, 2017M3A7B4049172, 2017R1A6A3A01012106, 2016R1D1A1B03933255, and 2018R1A5A6075964). This research was also supported by the Korea Institute of Science and Technology (2E28210 and 2E28190).

## Author contributions
D.C., M.-J.J., and J.-W.Y. designed the research. D.C. and M.-J.J. worked on device fabrication, characterization, and analysis. J.J., I.O., and J.P. assisted device fabrication and characterization. S.-I.K., H.-J.C., and S.-H.B. prepared LAO/STO samples. D.C., M.-J.J., and J.-W.Y. wrote the manuscript. H.-W.L., S.-M.H., H.J., H.C.K., B.-C.M., and S.-H.B. discussed the results and commented on the manuscript.

## Competing interests
The authors declare no competing interests.
