## [Peer Review File · Nature Communications]

Reviewers' comments:

Reviewer #1 (Remarks to the Author):

In this manuscript, authors studied the nonreciprocal charge transport in noncentrosymmetric oxide interfaces. They successfully observed nonreciprocal charge transport in the conductive oxide interface LaAlO₃/SrTiO₃, which shows the characteristic behavior for polar systems and found that it can be tuned and enhanced by gating. The results are impressive, clearly showing the potential of oxide interfaces. I believe that results are worth publishing in Nature Communications. However, I think the authors should further consider and clarify the following point before accepting the manuscript.

1. LaAlO₃/SrTiO₃ show the in-plane negative magnetoresistance as shown in Figs. 1 d and 2 a. What is the microscopic origin of it? Is it related with the out-of-plane magnetoresistance (weak (anti-)localization) discussed in Supplementary Fig. 5? What is the temperature dependence of the magnetoresistance? I also want to know the relation between the negative magnetoresistance and the sign of nonreciprocal charge transport.

2. Authors show the temperature dependence of $\Delta R_{xx}/R_{xx}$ in Fig. 2 c, which show the decrease below 10K. They attributed it to the decrease of conductivity in LaAlO₃/SrTiO₃. I think it is better to show the temperature dependence of ΔR_{xx} (raw data) in supplementary information.

3. In the inset of Fig.2 b, authors draw the schematic of electronic band structure of LaAlO₃/SrTiO₃. I want to know the typical quantitative value of the energy and carrier density of Lifshitz transition. Is it close to the present case ($n \sim 10^{14} \text{ cm}^{-3}$ according to the Supplementary Fig. 4)?

4. I agree that deviation from the B-linear behavior of $R_{2\omega}$ might come from the higher order terms. Why does it show increasing (not decreasing) behavior? Again, is it related with the negative magnetoresistance?

5. I am interested in the detailed field-angle dependence of nonreciprocal resistance. It seems that field-angle dependence for xy plane and that for zy plane are different. This means that signals of nonreciprocal charge transport deviate from the simple formula $\Delta R \propto I \cdot (P \times B)$. What is the potential reason? Is it related with the fact that LaAlO₃/SrTiO₃ interface is fourfold symmetrical?

6. In addition to Ref. 29 (Sci. Adv. 3, e1602390 (2017)), I recommend authors to cite two more related works which studied the enhanced nonlinear superconducting transport in noncentrosymmetric systems.

- F. Qin et al., Nat. Commun. 8, 14465 (2017).
- J. Lustikova et al., Nat. Commun. 9, 4922 (2018).

Reviewer #2 (Remarks to the Author):

NCOMMS-18-37975

Gate-tunable giant nonreciprocal charge transport in noncentrosymmetric oxide interfaces

D.Choe et al.

The work reports the appearance of non reciprocal charge transport, tunable by electric field effect, in the 2DEG at the LaAlO₃/SrTiO₃ interface. The authors perform DC and AC measurements of the resistance as a function of the direction of the injected current and report the behavior of the non reciprocal response as a function of gate voltage, magnetic field and angle between the current direction axis and magnetic field orientation. The magnitude of the non reciprocal response

found by the authors is larger than what has been found in many polar materials. They stress that this is due not only to the large Rashba spin-orbit interaction energy, but especially to the relative strength of the Rashba spin-splitting energy when compared to the Fermi energy.

The authors highlight the interesting point that the LAO/STO system shows many properties and, more interestingly, these are characterized by a special reciprocal energy scale.

The work is well written in the first part while the second, dealing with the AC measurements, is difficult to follow.

There are also some important issues which should be cleared by the authors.

- The authors write that “The strong asymmetric V_g dependence of the nonreciprocal response is a consequence of the V_g dependent Rashba spin-orbit interaction in combination with the n^{-3} dependence”. On the other hand, the authors observe a change in the carrier density from 1.65 to $1.9 \times 10^{13} \text{ cm}^{-2}$ (for $V_g=0$ and 200V respectively). This variation is weak, as also remarked by the authors, if compared with typical values found for LAO/STO 2DEG (see for instance A. Joshua et al., Nature Communications 3, 1129 (2012)). This seems to indicate that, although the carrier density does not change much, the Rashba spin-orbit interaction raises considerably with increasing gate voltage. The authors should provide more details on this issue. For instance, they could analyze the magnetoconductance curves (supplementary figure 5) to estimate the Rashba scattering parameters as a function of the gate voltage.

- The curves shown in Figure 1d and 2a remind the magnetoresistance hysteresis shown by Ayno et al. in Physical Review Materials 2, 031401(R) (2018) and attributed to a magnetothermal effect. The authors should provide evidence that this effect is not at play in their case. They should specify the field sweeping rate used. Moreover, high currents for both DC ($30\mu\text{A}$) and AC ($200\mu\text{A}$) measurements were used. Why did they chose such values? What happens if the bias current is reduced?

- The authors write that “The nonreciprocal response is nearly negligible for $V_g < 0 \text{ V}$, while it stiffly increases upon applying positive V_g (fig. 2b)”. On the other hand, the crossover between weak localization and weak anti-localization (indicating the increase in Rashba spin-orbit coupling) takes place at much lower gate voltages, between $V_g=-40\text{V}$ and $V_g=-80\text{V}$ (supplementary fig. 5). The authors should comment on this discrepancy.

- “The sheet carrier density (n_s) was $\sim 1.56 \times 10^{13} \text{ cm}^{-2}$ at 2 K ” From Supplementary figure 4, the minimum value reached by n_s seems to be $\sim 1.65 \times 10^{13} \text{ cm}^{-2}$

- In Figure 1 the panel showing the SEM picture of the device is missing

- The saturation of the nonreciprocal response at high V_g cannot be seen in Figure 2b.

In summary, the work could possibly be suitable for publication once the issues reported above have been addressed by the authors.

Reviewer: 1

Reviewer's Comments:

In this manuscript, authors studied the nonreciprocal charge transport in noncentrosymmetric oxide interfaces. They successfully observed nonreciprocal charge transport in the conductive oxide interface $\text{LaAlO}_3/\text{SrTiO}_3$, which shows the characteristic behavior for polar systems and found that it can be tuned and enhanced by gating. The results are impressive, clearly showing the potential of oxide interfaces. I believe that results are worth publishing in Nature Communications. However, I think the authors should further consider and clarify the following point before accepting the manuscript.

Our responses:

We greatly appreciate reviewer's a number of important comments and advices as well as positive remarks. Below is the detailed discussions and corrections we made in response to reviewer's comments.

#1. Reviewer's comments:

$\text{LaAlO}_3/\text{SrTiO}_3$ show the in-plane negative magnetoresistance as shown in Figs. 1 d and 2 a. What is the microscopic origin of it? Is it related with the out-of-plane magnetoresistance (weak (anti-)localization) discussed in Supplementary Fig. 5? What is the temperature dependence of the magnetoresistance? I also want to know the relation between the negative magnetoresistance and the sign of nonreciprocal charge transport.

Our responses:

We appreciate reviewer's comments for important issue, which we should have paid attention to.

- Large negative in-plane magnetoresistance (MR) in LAO/STO has been observed in several previous reports. In general, the observed negative in-plane MR in LAO/STO have been attributed to the presence of Kondo effect induced by magnetic impurities at the interface [Brinkman *et al. Nat. Mater.* **6**, 493-496 (2007); Ruhman *et al. Phys. Rev. B* **90**, 125123 (2014); Joshua *et al. PNAS* **110**, 9633 (2013)]. However, recent study of Diez *et al.* showed that the combination of spin-orbit coupling, band anisotropy, and finite-range impurity scatterings can explain the behavior of in-plane negative MR in LAO/STO [*Phys. Rev. Lett.* **115**, 016803 (2015)]. Here, the spin-orbit interaction along with Zeeman energy drives a highly anisotropic deformation of the Fermi surface, leading to suppressed interband scattering and reduced sheet resistance [*Phys. Rev. Lett.* **115**, 016803 (2015)].

In response to reviewer's comment, we added following sentence in the revised manuscript.

“This negative in-plane MR in LAO/STO was attributed to the anisotropic deformation of the Fermi surface upon increasing Zeeman energy, which results in suppressed interband scattering and reduced sheet resistance⁴⁰.”

- When the out-of-plane magnetic field was applied, quantum interference effect to the diffusive transport becomes significant, especially at low temperature. Thus, the mechanism of in-plane MR and out-of-plane MR are different, although both MR highly rely on the spin-orbit interaction.

In LAO/STO, a negative MR in response to the out-of-plane magnetic field can be observed for a large negative gate voltage ($-80 V_g$), indicating the dominance of weak localization due to weak spin-orbit interaction (Supplementary Fig. 5). As the applied gate voltage increased, a negative MR gradually turned into a positive MR, indicating that the charge transport relies on the weak anti-localization (Supplementary Fig. 5). On the other hand, when the in-plane field was applied, the negative MR get stronger as the applied gate voltage increased. Thus, the gate-dependent evolution of MR is opposite to each other between in-plane and out-of-plane MR. However, we cannot argue that both MRs (in-plane and out-of-plane) are independent each other, because both mechanisms highly rely on the strength of spin-orbit interaction.

- The in-plane negative MR shows maximum magnitude at around 8 K. It is progressively suppressed as the temperature is raised but still clearly visible at 20K, in agreement with previous report [*Phys. Rev. Lett.* **115**, 016803 (2015)]. In response to reviewer's comments, we added the temperature dependent in-plane MR curves in Supplementary Fig. S10.

- As we discussed, the sign of nonreciprocal charge transport depends on the direction of polarity and magnetic field as follows, $\Delta R = R(I) - R(-I) \propto I \cdot (P \times B)$. If the direction of magnetic field or polarity is reversed, ΔR changes sign. According to the explanation of Diez *et al.* (*Phys. Rev. Lett.* 115, 016803 (2015)), the negative in-plane MR arises from the suppressed interband scattering due to anisotropic deformation of the Fermi surface. Thus, the sign of negative in-plane MR does not change for the reversal of B_y or P_z . As the reviewer perceived, it seems that both mechanisms are highly correlated, because both effects depend on the strength of spin-orbit coupling.

In response to reviewer's comment, we added following sentence in the revised manuscript.

“Interestingly, the negative in-plane MR also increases significantly with applying positive V_g and can be collapsed into a single curve by a rescaling of the magnetic field $B \rightarrow B/B^*$ (B^* is a density dependent value)⁴⁰ (see Supplementary Fig. 9). Mechanisms of nonreciprocal charge transport and negative in-plane MR could be highly correlated because both effects depend on the anisotropic deformation of the Fermi surface”

#2. Reviewer's comments:

Authors show the temperature dependence of $\Delta R_{xx}/R_{xx}$ in Fig. 2 c, which show the decrease below 10K. They attributed it to the decrease of conductivity in LaAlO₃/SrTiO₃. I think it is better to show the temperature dependence of ΔR_{xx} (raw data) in supplementary information.

Our responses:

Following reviewer's suggestion, we have added the temperature dependence of ΔR_{xx} in Fig. 2c and in-plane MR curves (raw data) in supplementary Fig. 10. Below 10 K, nonreciprocal responses $\Delta R_{xx}/R_{xx}$ decreases with decreasing temperature. We believe that it would be associated with quantum interference effect, which get stronger below 10 K. Other possible explanation is the slight decrease of carrier concentration when the temperature is lowered below 10 K (see the inset of Supplementary Fig. 4b). The decrease of carrier concentration away from the Lifshitz transition reduces spin-orbit coupling, so does the nonreciprocal response.

In response to reviewer's valuable comment, we modified Fig. 2c and Supplementary Fig. 4b and we added Supplementary Fig. 10 and 11, and following sentences in the revised manuscript. "Another possible explanation is the slight decrease of a carrier concentration when the temperature is lowered below 10 K (Supplementary Fig. 4b). The decrease of a carrier concentration away from the Lifshitz transition reduces the spin-orbit coupling, so does the nonreciprocal response."

#3. Reviewer's comments:

In the inset of Fig.2 b, authors draw the schematic of electronic band structure of LaAlO₃/SrTiO₃. I want to know the typical quantitative value of the energy and carrier density of Lifshitz transition. Is it close to the present case ($n \sim 10^{14} \text{ cm}^{-2}$ according to the Supplementary Fig. 4)?

Our responses:

Thanks for pointing out important issue. According to literatures, typical quantitative values of the carrier density for Lifshitz transition were $1.68 \pm 0.18 \times 10^{13} \text{ cm}^{-2}$ by Joshua [Nat. Commun. 3, 1129 (2012)]. If we estimate Fermi energy based on 2D free electron model, this value of n corresponds to $\epsilon_F \sim 20.1 \text{ meV}$. In our study, the obtained value of n_s with zero gate voltage is $\sim 1.61 \times 10^{13} \text{ cm}^{-2}$ at 8 K and $1.56 \times 10^{13} \text{ cm}^{-2}$ at 2 K (for device B, Supplementary Fig. 4b). This value is slightly less than the Lifshitz transition. Thus, we observed strongly enhanced nonreciprocal response with increasing V_g across zero voltage. Note that there is slight sample to sample variation in n , so does gate voltage required to induce the Lifshitz transition.

In response to the reviewer's comments, we added following sentences in the revised manuscript. "According to the literature by Joshua *et al.*¹⁰, typical quantitative values of the carrier density for Lifshitz transition were $1.68 \pm 0.18 \times 10^{13} \text{ cm}^{-2}$. This value corresponds to the Fermi energy of

~ 20.1 meV within the 2D free electron model. Not that the obtained value of n_s with $V_g = 0$ is ~ $1.61 \times 10^{13} \text{ cm}^{-2}$ at 8 K (for device B, Supplementary Fig. 4). This value is slightly less than the Lifshitz transition. Thus, Lifshitz transition could occur with increasing V_g across zero voltage.”

#4. Reviewer's comments:

I agree that deviation from the B-linear behavior of $R_{2\omega}$ might come from the higher order terms. Why does it show increasing (not decreasing) behavior? Again, is it related with the negative magnetoresistance?

Our responses:

We thank for reviewer's positive remarks and constructive questions. The high order terms depend on the equation, as follow $J_x^{2\text{nd}} = E_x^2 (a \left(\frac{B_y}{\lambda}\right) + b \left(\frac{B_y}{\lambda}\right)^3 + O\left(\left(\frac{B_y}{\lambda}\right)^5\right))$. The appearance of higher order term as we discussed depends on the relative strength between magnetic field energy and Rashba spin-orbit splitting energy. Basically, nonreciprocal charge transport occurs due to the imbalance of Fermi momentum between leftward and rightward carriers. At $B_y \ll \lambda$, this imbalance linearly increases with increasing field. When $B_y \sim \lambda$, it strongly enhanced with increasing field. Thus, the high order dependences are reinforcing components to the nonreciprocal response rather than detrimental components.

- As we mentioned in our previous response, large negative in-plane MR is associated with the spin-orbit interaction along with Zeeman energy, which drives a highly anisotropic Fermi surface, leading to suppressed interband scattering and reduced sheet resistance. The anisotropic deformation of Fermi surface gets stronger with increasing magnetic field at high field regime, so does the negative in-plane MR. As reviewer perceived, it seems that both mechanisms are highly correlated, because both effects are contingent on the strength of spin-orbit coupling.

In response to reviewer's valuable comments, we added following sentences in the revised manuscript.

“Interestingly, the negative in-plane MR also increases significantly with applying positive V_g and can be collapsed into a single curve by a rescaling of the magnetic field $B \rightarrow B/B^*$ (B^* is a density dependent value)⁴⁰ (see Supplementary Fig. 9). Mechanisms of nonreciprocal charge transport and negative in-plane MR could be highly correlated because both effects depend on the anisotropic deformation of Fermi surface”

#5. Reviewer's comments:

I am interested in the detailed field-angle dependence of nonreciprocal resistance. It seems that field-angle dependence for xy plane and that for zy plane are different. This means that signals of

nonreciprocal charge transport deviate from the simple formula $\Delta R \propto I \cdot (P \times B)$. What is the potential reason? Is it related with the fact that LaAlO₃/SrTiO₃ interface is fourfold symmetrical?

Our responses:

We thank reviewer for bringing attention to the angle-dependent behavior of nonreciprocal charge transport. Similar to the angle dependent in the xy plane, the $R_{2\omega}$ is largest at $B \parallel y$ axis ($\theta = 90^\circ, 270^\circ$) for the zy plane. It is noticeable, however, that the $R_{2\omega}$ does not simply scale with B_y for the rotation in the zy plane. This different angle-dependent $R_{2\omega}$ for the rotation in the zy -plane is due to the higher-order dependence on magnetic field at high field regime. When the field is rotated in the xy -plane, the direction of field is always orthogonal to the direction of polarization (see figure below). Thus, the orthogonal component of B with respect to P is constant. Because $\Delta R \propto I \cdot (P \times B)$, component of I parallel to $(P \times B)$, which varies as $I \cdot \cos \theta$ during the field rotation in the xy plane, is subject to ΔR . Thus, we observe sinusoidal $R_{2\omega}$ for field rotation in the xy -plane as shown in Fig. 3b. On the other hand, when the field is rotated in the zy -plane, the component of field, which is orthogonal to the direction of polarization, is not constant and varies as $B \cdot \sin \theta$ (see figure below). Because the ΔR has additional higher order dependence on B at high magnetic field regime, the variation of ΔR become more significant at high field regime, making sharp increase of ΔR near 90 degree and 270 degree in the zy -plane. At relatively low magnetic field, where the ΔR is linear to the B , $R_{2\omega}$ displays sinusoidal behavior for the rotation of the magnetic field in the zy -plane (see Supplementary Fig. 18).

In response to reviewer’s valuable comment, we added following sentences in the revised manuscript.

“The higher order dependence on the applied magnetic field also reflects on the different behavior of $R_{2\omega}$ in between xy - and zy -plane rotations (shown in Fig. 3c). When the magnetic field is rotated in the xy -plane, the direction of the field is always orthogonal to the direction of the polarization. Because $\Delta R \propto I \cdot (P \times B)$, $R_{2\omega}$ displays sinusoidal behavior for the rotation of the magnetic field in the xy -plane. On the other hand, when the field is rotated in the zy -plane, the orthogonal component of the field to the direction of the polarization is not constant and

varies as $B \cdot \sin \theta$. Because the ΔR has additional higher order dependences on B at a high magnetic field regime, the variation of $R_{2\omega}$ becomes more significant at high fields, making sharp increase of $R_{2\omega}$ near 90° and 270° in the zy -plane (Fig. 3c). At a relatively low magnetic field, where ΔR is linear to B , $R_{2\omega}$ displays sinusoidal behavior for the rotation of the magnetic field in the zy -plane (see Supplementary Fig. 18).”

#6. Reviewer’s comments:

In addition to Ref. 29 (Sci. Adv. 3, e1602390 (2017)), I recommend authors to cite two more related works which studied the enhanced nonlinear superconducting transport in noncentrosymmetric systems.

- F. Qin et al., Nat. Commun. 8, 14465 (2017).
- J. Lustikova et al., Nat. Commun. 9, 4922 (2018).

Our responses:

We appreciate that the reviewer provided and reminded us for other related reports. We agree that those pioneering works should be discussed in our introduction. In response to reviewer’s comments, we added these references in the revised manuscript.

In short, the major changes we made in response to the reviewer’s comments are as follows.

1. Addition of analysis on the out-of-plane MR based on Maekawa-Fukuyama theory and Supplementary Note 1 and Fig. 5-7.
2. Addition of discussion on the negative in-plane MR and Supplementary Fig. 9-11.
3. Addition of discussion on the different $R_{2\omega}$ behavior in zy -plane and Supplementary Fig. 18.
4. Further discussion on the carrier concentration of the Lifshitz transition.
5. Modification of the description on the V_g dependent nonreciprocal resistance.
6. Further discussion on the decrease of nonreciprocal response below 10 K.
7. Modification of the description on the AC measurement results.

Reviewer: 2

Reviewer's Comments:

The work reports the appearance of nonreciprocal charge transport, tunable by electric field effect, in the 2DEG at the LaAlO₃/SrTiO₃ interface. The authors perform DC and AC measurements of the resistance as a function of the direction of the injected current and report the behavior of the nonreciprocal response as a function of gate voltage, magnetic field and angle between the current direction axis and magnetic field orientation. The magnitude of the nonreciprocal response found by the authors is larger than what has been found in many polar materials. They stress that this is due not only to the large Rashba spin-orbit interaction energy, but especially to the relative strength of the Rashba spin-splitting energy when compared to the Fermi energy.

The authors highlight the interesting point that the LAO/STO system shows many properties and, more interestingly, these are characterized by a special reciprocal energy scale.

The work is well written in the first part while the second, dealing with the AC measurements, is difficult to follow.

There are also some important issues which should be cleared by the authors.

Our responses:

We appreciate reviewer's a number of important comments and advices as well as positive remarks. In response to reviewer's comments, we modified part of AC measurements to improve legibility for general audiences. Thanks to reviewer's comments, our revised manuscript has been improved significantly. Below is the detailed discussions and corrections we made in response to reviewer's comments.

#1. Reviewer's comments:

The authors write that "The strong asymmetric V_g dependence of the nonreciprocal response is a consequence of the V_g dependent Rashba spin-orbit interaction in combination with the n^{-3} dependence". On the other hand, the authors observe a change in the carrier density from 1.65 to $1.9 \times 10^{13} \text{ cm}^{-2}$ (for $V_g=0$ and 200 V respectively). This variation is weak, as also remarked by the authors, if compared with typical values found for LAO/STO 2DEG (see for instance A. Joshua et al., Nature Communications 3, 1129 (2012)). This seems to indicate that, although the carrier density does not change much, the Rashba spin-orbit interaction raises considerably with increasing gate voltage. The authors should provide more details on this issue. For instance, they could analyze the magnetoconductance curves (supplementary figure 5) to estimate the Rashba scattering parameters as a function of the gate voltage.

Our responses:

We greatly appreciate reviewer's important comments. We analyzed the magnetoconductance curves as a function of the gate voltage. We obtained the Rashba spin-orbit interaction according to Maekawa-Fukuyama (MF) form as studied by Caviglia et al. [*Phys. Rev. Lett.* **104**, 126803 (2010)]. Results shows the sharp increase of the spin-orbit coupling constant α as we move across the quantum critical point and the corresponding rise of the spin splitting Δ (Supplementary Fig. 7). The V_g dependent variation of spin splitting Δ was more significant than that of the carrier density. Therefore, as reviewer emphasized, gate-tuned Rashba interaction mainly accounts for the observed V_g dependence of the nonreciprocal response in this system.

In response to reviewer's comments, we added Supplementary Fig. 6 and 7 and Note 1 and following sentence in the revised manuscript.

“Further analyses of out-of-plane MR curves within a Maekawa-Fukuyama theory were discussed in Supplementary Note 1 and Fig. 6 and 7. Result showed that the strong enhancement of the Rashba spin-orbit interaction across the Lifshitz point (Supplementary Fig. 7a).”

“The carrier concentration of the studied LAO/STO system exhibits gradual increase with increasing gate voltage but its variation is very weak (Supplementary Fig. 7c and 8). In contrast, the estimated Rashba spin splitting energy is significantly enhanced with increasing V_g (Supplementary Fig. 7a). Therefore, gate-tuned Rashba interaction mainly accounts for the observed V_g dependence of the nonreciprocal response in this system.”

#2. Reviewer's comments:

The curves shown in Figure 1d and 2a remind the magnetoresistance hysteresis shown by Ayno et al. in *Physical Review Materials* 2, 031401(R) (2018) and attributed to a magnetothermal effect. The authors should provide evidence that this effect is not at play in their case. They should specify the field sweeping rate used. Moreover, high currents for both DC (30 μ A) and AC (200 μ A) measurements were used. Why did they choose such values? What happens if the bias current is reduced?

Our responses:

We agree with reviewer's important concern and appreciate important comments. The paper from Ayno [*Phys. Rev. Mater.* 2, 031401(R) (2018)] showed the magnetoresistance (MR) hysteresis induced by a magnetothermal effect. The magnetoresistance hysteresis is the resistance difference between + to - field sweep and - to + field sweep measurements. The nonreciprocal charge transport in our report is the resistance difference between measurements with +I and -I currents. Thus, we are dealing with different phenomena.

-In the report of Ayno et al, the MR hysteresis appears in the vicinity of the small magnetic field and disappear by increasing the magnetic field over 1T. This behavior occurs when magnetic anisotropy energy overcome the thermal effect. Thus, it occurs when the temperature was

lowered below ~ 800 mK. In our study, the nonreciprocal responses persist over several tenth Kelvin. And it is negligible in the vicinity of the small magnetic field, and increases linearly with increasing field, and finally it diverges with higher order dependence at high magnetic field. Therefore, the overall behavior of the studied nonreciprocal response is completely different and is not related with magnetothermal effect. The MR measurement was done with magnetic field sweeping rate of 10 mT/s.

- Our measurement was typically done with the current of DC (30 μ A) and AC (200 μ A) to clearly observe the nonreciprocal response. As can be seen in Figure 4d, the nonreciprocal response linearly proportional to the electric current as $\Delta R \propto I \cdot (P \times B)$. As the reviewer commented, if the bias current is reduced, the nonreciprocal response linearly decreases. Applying high current may introduce deviation from linear relationship due to the heating effect. Thus, we chose moderate high current of DC (30 μ A) and AC (200 μ A) to obtain clear enough signal to noise ratio.

In response to reviewer's comments, we added following sentences.

In the revised manuscript,

“Measurement were done with magnetic field sweeping rate of 10 mT/s.”

In Supplementary Fig. 11,

“We note that the observed ΔR_{xx} is not associated with magnetothermal effect, which appears in the vicinity of the small magnetic field at very low temperature ($< \sim 800$ mK) [Phys. Rev. Mater. 2, 031401(R) (2018)].”

#3. Reviewer's comments:

The authors write that “The nonreciprocal response is nearly negligible for $V_g < 0$ V, while it stiffly increases upon applying positive V_g (fig. 2b)”. On the other hand, the crossover between weak localization and weak anti-localization (indicating the increase in Rashba spin-orbit coupling) takes place at much lower gate voltages, between $V_g = -40$ V and $V_g = -80$ V (supplementary fig. 5). The authors should comment on this discrepancy.

Our responses:

We appreciate the reviewer's careful proofreading and valuable comments. The crossover between weak localization and weak anti-localization are associated with relative scale between phase coherence length and spin diffusion length. Thus, the crossover between weak localization and weak anti-localization may occur before the Lifshitz transition, where the spin-orbit coupling starts to increase more strongly. As reviewer suggested, we estimated spin-orbit splitting by fitting with Maekawa-Fukuyama (MF) theory to the out-of-plane MR. Results show that even before the Lifshitz transition, spin-orbit coupling slightly increases with increasing V_g (Supplementary Fig. 7a). Thus, the change of relative scale between phase coherence length and spin diffusion length may occurs before the Lifshitz transition (Supplementary Fig. 7c). In

contrast, the nonreciprocal responses would directly rely on the strength of spin-orbit coupling as follows $\Delta R \propto I \cdot (P \times B)$, where P is the polarization. Here, the polarization P is directly associated with the Rashba spin-orbit coupling, which is proportional to the potential gradient. We also note that there is slight sample to sample variation in n_s , so does gate voltage required to induce Lifshitz transition and transition between weak localization and weak anti-localization.

In response to the reviewer's comments, we added further discussions and analysis on the weak-localization and weak-antilocalization and discrepancy with the Lifshitz transition in the Supplementary Note 1 and Fig. 6 and 7.

#4. Reviewer's comments:

“The sheet carrier density (n_s) was $\sim 1.56 \times 10^{13} \text{ cm}^{-2}$ at 2 K”. From Supplementary figure 4, the minimum value reached by n_s seems to be $\sim 1.65 \times 10^{13} \text{ cm}^{-2}$.

Our responses:

Thanks for careful proofreading. According to the Supplementary Fig. 4, the estimated value of n_s was $\sim 1.61 \times 10^{13} \text{ cm}^{-2}$ at 8 K and $1.56 \times 10^{13} \text{ cm}^{-2}$ at 2 K, which was estimated from device B. The results in the main text were obtained from device A. The estimated sheet carrier density (n_s) were nearly identical between device A and device B. For device A, the estimated n_s was $\sim 1.56 \times 10^{13} \text{ cm}^{-2}$ at 2 K. Supplementary Fig. 8 shows the estimated n_s upon varying V_g measured for device D. In this case, the estimated n_s was $\sim 1.65 \times 10^{13} \text{ cm}^{-2}$ at 8 K and $V_g = 0$. We also note that there is slight sample to sample variation in n_s , so does gate voltage required to induce Lifshitz transition.

In response to the reviewer's comments, we added inset in Supplementary Fig. 4b, which clearly displays values of carrier density at low temperatures.

#5. Reviewer's comments:

In Figure 1 the panel showing the SEM picture of the device is missing.

Our responses:

Fig. 1c in the revised manuscript displays the SEM image of the studied device A and we improved contrast of this figure.

#6. Reviewer's comments:

The saturation of the nonreciprocal response at high V_g cannot be seen in Figure 2b.

Our responses:

We thank for the reviewer's careful proofreading. As can be seen in Fig. 2b, the nonreciprocal response significantly increases with increasing V_g across 0 V. But its enhancement become less effective when V_g increases further. As reviewer pointed out, we didn't observe the saturation. But overall behavior appears to show the reduced enhancement with increasing V_g further.

In response to reviewer's comments, we added following sentence in the revised manuscript. "Therefore, gate-controlled Rashba interaction mainly accounts for the observed V_g dependence of the nonreciprocal response in this system. As shown in Fig. 2b, the observed nonreciprocal response is nearly negligible for $V_g < 0$ V, while it stiffly increases upon applying positive V_g , in consistent with the Lifshitz transition across zero gate voltage."

In short, the major changes we made in response to the reviewer's comments are as follows.

1. Addition of analysis on the out-of-plane MR based on Maekawa-Fukuyama theory and Supplementary Note 1 and Fig. 5-7.
2. Addition of discussion on the negative in-plane MR and Supplementary Fig. 9-11.
3. Addition of discussion on the different $R_{2\omega}$ behavior in zy -plane and Supplementary Fig. 18.
4. Further discussion on the carrier concentration of the Lifshitz transition.
5. Modification of the description on the V_g dependent nonreciprocal resistance.
6. Further discussion on the decrease of nonreciprocal response below 10 K.
7. Modification of the description on the AC measurement results.

Reviewers' comments:

Reviewer #1 (Remarks to the Author):

Authors answered all the questions appropriately and manuscript is now suitably revised. Although I believe that it can be ready for acceptance, I have another minor question after reading the response to comment 3 (typical quantitative value of the energy and carrier density of Lifshitz transition). According to the authors, carrier density of their sample is slightly less than the Lifshitz transition and Lifshitz transition could occur with increasing V_g across zero voltage. Is there any signature (or anomaly) reflecting the Lifshitz transition in $R\omega$ or $R2\omega$? I think it may be an important future issue.

Reviewer #2 (Remarks to the Author):

In this revised version, the authors included the analysis of the magnetoconductance (MC) curves as a function of the gate voltage. This analysis is important to reply to some questions both the First Reviewer and I asked during the first review stage. However, the data added and their analysis raise many doubts, in my mind.

1. In the Maekawa and Fukuyama formula, the minimum of the differential MC curve gives an estimation of the B_{so} field (W. Knapp et al., Phys. Rev B 53, 3912 (1996)). Thus, for B_{so} fields in the order of 1T (as is typically found in LAO/STO devices) the MC data should be taken up to several Tesla, in order to have a clear picture of the evolution of the curve minimum. In Supplementary Fig. 6, on the other hand, the authors show MC data up to 1.2T. This is too low to obtain a reliable fit of the curves.

2. Supplementary Fig. 7b shows that B_i changes of more than one order of magnitude upon application of the gate voltage. On the other hand, in many other reports of LAO/STO magnetotransport, B_i is found to change little with gate voltage (see Physical Review Letters 104, 126803 (2010), Physical Review B 90, 235426 (2014), Scientific Reports 5, 12751 (2015)). The two different behaviors could be reconciled if, in the present manuscript, the tuning of the carrier concentration was large. However, this is not the case, since in the gate voltage range explored, the carrier concentration change only from 1.5 to $1.9 \times 10^{13} \text{cm}^{-2}$. The authors should check carefully the consistence of their results with the wide literature available.

3. To extract the data shown in Supplementary Fig. 7 the authors use an effective mass $m^* = 2m_e$. However, the effective mass of charge carriers in LAO/STO changes across the Lifshitz transition. In Physical Review B 86, 201105(R) (2012) a change from $0.7m_e$ to $2.2m_e$ when going from d_{xy} to $d_{\{xz,yz\}}$ dominated transport, was calculated. This change would modify the values of the diffusion constant D , of the scattering times and above all of the Rashba spin-orbit interaction constant α shown by the authors in Suppl. Fig. 7.

The increase the Rashba spin-orbit in LAO/STO 2DEG across the Lifshitz transition has been reported by several other authors. I am not challenging the occurrence of such phenomena but the data analysis used by the authors to support this idea in their work.

In conclusion, as I wrote in the previous report, this work is interesting. However, some of the affirmation it contains seem to be founded on approximate and partial analysis. Even though some of these analysis are not part of the main message, they are required to support it.

I think that a manuscript suitable for publication in an high impact journal should contain only data analysis performed with great accuracy and precision. Therefore I cannot recommend this manuscript for publication in Nature Communications.

Reviewer: 1

Reviewer's Comments:

Authors answered all the questions appropriately and manuscript is now suitably revised. Although I believe that it can be ready for acceptance, I have another minor question after reading the response to comment 3 (typical quantitative value of the energy and carrier density of Lifshitz transition). According to the authors, carrier density of their sample is slightly less than the Lifshitz transition and Lifshitz transition could occur with increasing V_g across zero voltage. Is there any signature (or anomaly) reflecting the Lifshitz transition in R_{ω} or $R_{2\omega}$? I think it may be an important future issue.

Our responses:

We greatly appreciate reviewer's positive remarks and important comments. In addition to the direct measurement of band structure, the signature of Lifshitz transition can also be evidenced through various analysis of transport properties. For example, the SdH oscillation allow us to investigate the Fermi surface. In this specific system of LAO/STO, Lifshitz transition closely associated with the strength of Rashba spin-orbit interaction. Thus, the analysis of out-of-plane magnetoresistance curve based on MF theory could exhibit abrupt increase of spin-orbit interaction at the Lifshitz point, as shown in supplementary Figure 7. These analysis requires multiple fitting procedure with many parameters. For ac measurement, the linear components of R_{ω} upon applying out-of-plane field could be used for such time-consuming analysis to find out Lifshitz point. On the other hand, the nonreciprocal $R_{2\omega}$ is a physical property that is directly associated with the strength of spin-orbit interaction and can be utilized for the estimation of the size of Rashba constant. Thus, sudden increase of $R_{2\omega}$ can be regarded as the signature of the Lifshitz point in LAO/STO system.

Another signature of Lifshitz transition in LAO/STO is the change of symmetry in angular dependence of magnetoresistance, as done in PNAS 110, 9633-9638 (2013). We agree that it is an interesting issue for future work. At this point, we observed that the symmetry of angular dependent $R_{2\omega}$ do not vary across the Lifshitz transition and only the magnitude of $R_{2\omega}$ increases steeply across the transition, as shown in Fig. 4a.

In response to the reviewer's comments, we added following sentence in the revised manuscript. "As the nonreciprocal response $R_{2\omega}$ is directly associated with the strength of the Rashba spin-orbit interaction, the sudden increase of $R_{2\omega}$ could be regarded as the signature of the Lifshitz transition in this LAO/STO system."

Reviewer: 2

Reviewer's Comments:

In this revised version, the authors included the analysis of the magnetoconductance (MC) curves as a function of the gate voltage. This analysis is important to reply to some questions both the First Reviewer and I asked during the first review stage. However, the data added and their analysis raise many doubts, in my mind.

Our responses:

We greatly appreciate reviewer's careful proofreading and a number of valuable comments. We agree that we should have paid more attention on the analysis of magnetoconductance (MC) to produce high-precision analysis and to improve the completeness of our manuscript. In order to produce more reliable analysis, we performed additional experiment with newly fabricated device with further caution to get more reliable results, which allowed us to get more precise fitting analysis on MC. Thanks to reviewer's important comments, our analyses on MC have been significantly improved and the obtained results became more reliable and consistent with previous reports on MC.

#1. Reviewer's comments:

In the Maekawa and Fukuyama formula, the minimum of the differential MC curve gives an estimation of the B_{so} field (W. Knapp et al., Phys. Rev B 53, 3912 (1996)).

Thus, for B_{so} fields in the order of 1T (as is typically found in LAO/STO devices) the MC data should be taken up to several Tesla, in order to have a clear picture of the evolution of the curve minimum. In Supplementary Fig. 6, on the other hand, the authors show MC data up to 1.2T. This is too low to obtain a reliable fit of the curves.

Our responses:

We appreciate reviewer for reminding us of valuable literature on WL/WAL. We have performed additional experiment for MC measurement with newly fabricated device in order to get more robust MC results with reduced signal to noise. We have extended the range of fitting up to 4 T to cover the minimum of the differential MC curves (Supplementary Fig. 6 in revised manuscript). We also noticed that the fitting to MC results start to deviate at high field as the V_g was increased far above from the Lifshitz point due to orbital magnetoresistance. This behavior is also reported in previous reports [Nat. Comm. **6**, 6028 (2015)]. Thus, our analysis is primary focused on the range of V_g around Lifshitz point, where we could obtain excellent fitting, as shown in Supplementary Fig. 6.

#2. Reviewer's comments:

Supplementary Fig. 7b shows that B_i changes of more than one order of magnitude upon application of the gate voltage. On the other hand, in many other reports of LAO/STO magnetotransport, B_i is found to change little with gate voltage (see Physical Review Letters 104, 126803 (2010), Physical Review B 90, 235426 (2014), Scientific Reports 5, 12751 (2015)). The two different behaviors could be reconciled if, in the present manuscript, the tuning of the carrier concentration was large. However, this is not the case, since in the gate voltage range explored, the carrier concentration change only from 1.5 to $1.9 \times 10^{13} \text{cm}^{-2}$.

The authors should check carefully the consistence of their results with the wide literature available.

Our responses:

We appreciate reviewer for careful proof reading. In our previous result, the variation of B_i was negligible until the V_g was increased far above the Lifshitz point. As we mentioned, the fitting to MC result start to deviate at high field as the V_g was increased far above from the Lifshitz point. This behavior was also reported in previous reports [Nat. Comm. 6, 6028 (2015)]. Thus, our analysis in the revised manuscript is primary focused on the range of V_g around Lifshitz point. By using the improved fitting with extended region of magnetic field, our analysis show that the change of B_i was not significant upon varying V_g , as shown in Supplementary Fig. 7b.

#3. Reviewer's comments:

To extract the data shown in Supplementary Fig. 7 the authors use an effective mass $m^* = 2m_e$. However, the effective mass of charge carriers in LAO/STO changes across the Lifshitz transition. In Physical Review B 86, 201105(R) (2012) a change from $0.7m_e$ to $2.2m_e$ when going from d_{xy} to $d_{\{xz,yz\}}$ dominated transport, was calculated. This change would modify the values of the diffusion constant D , of the scattering times and above all of the Rashba spin-orbit interaction constant α shown by the authors in Suppl. Fig. 7.

Our responses:

We thanks reviewer for reminding us additional analysis. The ref. of Physical Review B 86, 201105(R) (2012) (ref. [15] in the manuscript) estimated effective mass by rewriting the relation of $B_{so} = \hbar/4eD\tau_{so}$ with an assumption of Dyakonov-Perel spin relaxation, $D\tau_{so} = \frac{1}{2}v_F^2 \frac{2\pi}{\Omega_{so}^2}$ (where $m^*v_F = \hbar k_F$ and $\Omega_{so} = \frac{2\alpha k_F}{\hbar}$). Then, the m^* can be expressed as follows

$$m^* = \frac{\hbar^2}{4\pi\alpha} \sqrt{\frac{B_{so}}{\Phi_0}},$$

where $\Phi_0 = \frac{h}{2e}$ and α is the strength of Rashba spin-orbit interaction. Here, they assumed that the Rashba spin-orbit interaction has a linear dependence on the V_g , *i.e.* $\alpha = \lambda E$, in their wedge model (λ is the material specific Rashba spin-orbit coefficient, E is the electric field in the quantum well).

We also followed the analysis done in ref [15] and added results in Supplementary Fig. 8. Our results show that the charge transport evolving from being d_{xy} dominated ($m^* \sim 0.62m_e$) to being d_{xz}, d_{yz} dominated ($m^* \sim 2.3m_e$) across the Lifshitz transition, in consistent with previous report.

As the reviewer mentioned, the change of effective mass modifies the values of the diffusion constant D , the scattering times, and the Rashba spin-orbit interaction constant α . If we solve for Rashba spin-orbit interaction constant $\alpha = \frac{\hbar^2}{2m^*} \sqrt{\frac{\pi}{D\tau_{so}}}$ with the calculated effective mass $m^* = \frac{\hbar^2}{4\pi\lambda E} \sqrt{\frac{B_{so}}{\Phi_0}}$ following the ref [15], then it simply reduce to the $\alpha = \lambda E$, which they assumed as a linear function to get the effective mass.

Therefore, if we assume fixed effective mass, we could obtain sudden increase of Rashba spin-orbit interaction across the Lifshitz transition. If we assume linear variation of Rashba spin-orbit interaction, we could obtain sudden increase of effective mass across the Lifshitz transition.

In short, we performed the analysis on MC results obtained from newly fabricated device based on MF theory. Further analysis on the other parameters, such as τ_{so} and α was obtained with fixed electron mass following the ref. [13] (Supplementary Fig. 7). In response to reviewer's comments, we also performed estimation of effective mass based on the assumption of linear variation of Rashba spin-orbit interaction following the ref. [15] (Supplementary Fig. 8). Additional references of earlier studies (ref. 13, 15, 41, 42 in the revised manuscript) on WL/WAL in LAO/STO was also included in the revised manuscript.

REVIEWERS' COMMENTS:

Reviewer #1 (Remarks to the Author):

Authors answered my question adequately and manuscript is now suitably revised. I believe that it can be ready for acceptance.

Reviewer #2 (Remarks to the Author):

The authors performed additional magnetoconductance measurements and a more careful analysis of the data, obtaining results which reinforce their main message. I believe that now the manuscript meets the standards of accuracy required by an high impact factor journal, therefore I recommend publication in Nature Communications.